# Phytochemicals: A Promising Alternative for the Prevention of Alzheimer’s Disease

**DOI:** 10.3390/life13040999

**Published:** 2023-04-12

**Authors:** Bhupendra Koul, Usma Farooq, Dhananjay Yadav, Minseok Song

**Affiliations:** 1Department of Biotechnology, Lovely Professional University, Phagwara 144411, Punjab, India; 2Department of Botany, Lovely Professional University, Phagwara 144411, Punjab, India; 3Department of Life Sciences, Yeungnam University, Gyeongsan 38541, Republic of Korea

**Keywords:** Alzheimer’s disease, medicinal plants, phytochemicals, neurological diseases treatment

## Abstract

Alzheimer’s disease (AD) is a neurological condition that worsens with ageing and affects memory and cognitive function. Presently more than 55 million individuals are affected by AD all over the world, and it is a leading cause of death in old age. The main purpose of this paper is to review the phytochemical constituents of different plants that are used for the treatment of AD. A thorough and organized review of the existing literature was conducted, and the data under the different sections were found using a computerized bibliographic search through the use of databases such as PubMed, Web of Science, Google Scholar, Scopus, CAB Abstracts, MEDLINE, EMBASE, INMEDPLAN, NATTS, and numerous other websites. Around 360 papers were screened, and, out of that, 258 papers were selected on the basis of keywords and relevant information that needed to be included in this review. A total of 55 plants belonging to different families have been reported to possess different bioactive compounds (galantamine, curcumin, silymarin, and many more) that play a significant role in the treatment of AD. These plants possess anti-inflammatory, antioxidant, anticholinesterase, and anti-amyloid properties and are safe for consumption. This paper focuses on the taxonomic details of the plants, the mode of action of their phytochemicals, their safety, future prospects, limitations, and sustainability criteria for the effective treatment of AD.

## 1. Introduction

Alzheimer’s disease (AD) is a neurological disorder in humans caused by complex pathophysiological mechanisms that lead to loss of memory and cognition, death of neurons, loss of synapses, and damage of the brain, which culminates in death [1,2]. It is the world’s most prevalent form of dementia and the dominant neurodegenerative disorder. Neurodegenerative diseases signify a great public health alarm and social and economic problem due to the high death rates and high treatment costs. Additionally, current treatments for AD are only able to lessen its symptoms and cannot halt the progression of neurodegeneration [2]. Alois Alzheimer, a German psychiatrist (1864–1915), gave the first explanation of it in 1906. Henceforth, the condition was given the term “Alzheimer’s disease” by Kraepelin [3]. About half of the world’s population above the age of 80 years are suffering from AD which has been ranked as the seventh leading cause of death [4]. Presently, more than 55 million AD patients are reported across the globe, and it is predicted that number may double every five years, so as to reach 115 million by 2050 [4,5]. The two primary pathological symptoms of AD are considered to be senile plaques and neurofibrillary tangles. In the ageing brain, misfolded protein accumulates and leads to metabolic loss, oxidative stress-induced damage, and synapse dysfunction. In AD, oxidative damage is indicated by high levels of DNA oxidation products such as 8-hydroxydeoxyguanosine in the brain cell nucleus and mitochondria [6,7,8].

AD is divided into five stages, mild cognitive impairment (MCI), mild, moderate, severe, and very severe AD [4]. According to several studies, the very early stage is called MCI, which can last for years without changing and is primarily associated with memory loss as well as cognitive impairments. The mild AD stage is characterised by forgetfulness and short-term memory loss, loss of interest in hobbies, repetitive questioning, and change of routine. Patients may become unable to execute a variety of duties on their own as the condition progresses, especially in those tasks that require cognition [9,10]. In moderate AD, more abrupt shifts and impaired routine are seen along with early-stage psychological dementia, which follows ongoing cognitive deficits and care-related transitions. At this point, 30% of patients may also experience illusionary misidentifications, which is long-term memory loss [9]. Severe AD, the fourth stage, is marked by disturbed and restless sleep patterns, rising indications of psychological disorders associated with dementia, and may even require assistance in bathing, feeding, or dressing [11]. Hence, these patients are totally dependent on caretakers. The most advanced stage of AD is referred to as very severe AD and is characterised by limited verbal speech, such as the use of single words or short sentences that finally lead to no speaking, bed rest, and the loss of fundamental psychomotor abilities. Patients eventually lose the ability to perform any task independently. In addition to AD, at this stage, conditions such as pneumonia or ulcers may also cause death [12].

There are different mechanisms of neurodegeneration such as inflammation of neurons, oxidative stress, environmental conditions, genetics, aggregation of Aβ (β-amyloid) in the brain, and dysfunction of the mitochondria [13,14,15]. There is currently no known cure for AD, however, preventative strategies are a hot topic of discussion. Currently, there is a low rate of clinical development for AD medications, and the majority of medical research is focused on slowing the progression rather than treating patients [16]. Very few drugs such as donepezil, rivastigmine, galantamine, and memantine are approved by the FDA for the treatment of AD [17,18]. These drugs only manage the early symptoms, show various side effects, and are costly. Each year, more than $600 billion is spent globally on the treatment of AD [19]. Therefore, it is crucial to look for novel approaches for the treatment of AD [20,21] and studies have suggested that phytochemicals such as huperzine, galantamine, quercetin, resveratrol, rosmarinic acid, and many more are obtained from various plants and have the potential to safely reduce the risk of AD [22,23,24].

Unfortunately, explicit information regarding the use of phytochemicals in the treatment of AD and AD-related symptoms is fragmentary. We intend to bridge this gap and provide information regarding the same. This review focuses on the use of different phytochemicals and their clinical trials for the prevention of AD, their limitations, and future prospects.

## 2. Factors Contributing to AD

AD is thought to be a complex disorder with a number of risk factors (Figure 1) such as age, consumption of alcohol, smoking, poor diet, sedentary lifestyle, environmental factors, and certain health issues such as damage to the vascular system, dysfunctioning of the immune system, high blood pressure, diabetes, atherosclerosis etc., [25,26].

### 2.1. Age Factors

Considering that several pathological alterations in AD are similar to those seen during ageing, with the exception of their intensity, it has been suggested that AD may represent an accelerated form of ageing. Therefore, in AD there is an age-related reduction in brain weight and volume, ventricle widening, and dendritic and synaptic loss in specific parts of the cognitively intact brain [27]. There may be two additional ageing processes playing a role in AD. First, a myelin breakdown brought by ageing [28] and, the second is the damage of locus caeruleus cells (LC), which induces microglia to reduce Aβ production and transmit noradrenaline via terminal varicosities to the cortex [29]. Early occurrence of tau-immunoreactive tumor necrosis factor in the LC is linked with AD [30]. This suggests that vascular factors could contribute to Alzheimer’s illness. The blood–brain barrier may deteriorate with ageing due to cell death in the LC.

### 2.2. Degeneration of Anatomical Pathways

This is another hypothesis regarding the pathology of AD. These pathways generally link action to perception. Under these two pathways are included.

#### 2.2.1. Cholinergic Pathway

One of the initial hypotheses for the origin of AD was a particular degradation of the cholinergic neurotransmitter system. Many studies have shown the loss of acetylcholine in the AD brain. Further investigations revealed decreased levels of choline acetyltransferase (CAT) [31]. In addition, it is noted that neurons are lost in the nucleus basalis of the Meynert (nBM), a region of the brain that receives diffuse cholinergic projection [32]. Several researchers discovered elevated 5-hydroxytryptamine (5-HT) turnover in AD and hypothesised that this was caused by a selective loss of cortical 5-HT neurons [33].

#### 2.2.2. Cortico-Cortical Pathways

Multiple lines of evidence suggest that AD is characterized by the degradation of anatomical networks connecting various regions of the cerebral cortex [34,35]. The duplicated local neuronal circuit represented by “columns” or “modules” is a key aspect of the cerebral cortex’s architectural structure. Numerous studies indicate a connection between AD and the deterioration of these cortical circuits. Many researchers identified a decrease in synaptophysin reactivity in the cortex associated with AD and linked it to synapse loss in the temporal lobe [36]. Some studies discovered a reduction in the synaptic marker SP6 in every area of the AD brain [36,37].

### 2.3. Environmental Factors

There are several environmental factors linked to AD, although the majority of research focuses on three of them: exposure to aluminium (Al), the impact of head injury, and the influence of food and malnutrition [38].

#### 2.3.1. Aluminium Toxicity

The evidence that aluminium causes AD is somewhat circumstantial and argumentative. Additionally, out of 13 studies that examined aluminium levels in gross brain tissue, 9/13 discovered elevated levels in AD, compared to the controls [39], whereas 4/13 showed no significant differences. Since injured brains accumulate aluminium, therefore, the chances of AD increase. Pyramidal neurons that contain neurofibrillary tangle (NFT) have also been found to possess Al. It may bind to DNA and modify cytoskeletal proteins, causing NFT to develop. Apart from establishing that Al can be neurotoxic in acute dosages, neuropsychological diseases are caused by intense exposure to Al, for example, bladder irrigation which is used to treat haemorrhagic cystitis with a 1% alum solution, or dialysis dementia [40,41].

#### 2.3.2. Head Injury

Primary head trauma from injury extends to initially unaffected areas via inflammatory cytokines and worsens the initial injury as a result of the immune and microglial cell activation in the CNS [42]. Several findings pointed towards the connection between AD and head trauma. Amyloid precursor protein (APP) is present around Aβ deposits in neuronal perikarya. These results imply that one mechanism by which AD originates and spreads in the brain through cell-to-cell transfer is the brain injury-mediated formation of pathogenic proteins [43,44].

#### 2.3.3. Food and Malnutrition

Studies on how diet affects AD have been more prevalent in recent years. Antioxidants, vitamins, polyphenols etc. are a few dietary supplements that have been shown to reduce the risk of AD, although high-calorie foods and saturated fats exacerbate the risk of AD [45]. As a consequence of the nonenzymatic glycation of free amino groups in proteins, lipids, and nucleic acids, heat-sensitive micronutrients such as vitamin C and folates degrade, a significant quantity of water is lost, and hazardous secondary products (advanced glycation end products: AGEs) are formed [46,47]. The harmful impact of AGEs is their capacity to cause oxidative stress and inflammation by altering the structure and function of the body’s proteins and cell surface receptors. Several investigations have shown that a raised AGEs serum level is linked to cognitive deterioration and the development of AD [48]. The AGE receptor (RAGE), which is present in the microglia and astrocytes among other body tissues, has been shown to be overexpressed in the brains of AD patients where it serves as a cell surface receptor and a transporter for Aβ [49]. An additional AD risk factor is malnutrition, as low levels of minerals, including vitamin D, vitamin B12, and folate, may impair cognitive performance. Patients with AD also face difficulties swallowing and eating, which raises the chances of malnutrition [50].

### 2.4. Genetic Factors

There is proof that first-degree relatives of AD patients have a higher probability of developing AD. A few genes (APP, PSEN, and Apo E) have been associated with AD [50].

#### 2.4.1. Amyloid Precursor Protein (APP)

The APP gene which is located on chromosome 21 codes for a type I transmembrane protein that is cleaved by the enzymes α-, β-, and γ-secretase to release Aβ and other proteins. The APP gene has thirty mutations, of which twenty-five are linked to AD and result in increased quantities of Aβ. Strong evidence links some uncommon variants of early-onset familial Alzheimer’s disease (EO-FAD) to particular genetic variables and a few cases of AD seen during the early 1990s were connected to APP gene alterations [51,52]. The most prevalent Aβ-peptide formed by the cleavage of APP is Aβ-42, which is mostly seen in senile plaque (SP). The greater soluble Aβ-40 is also found in connection with cerebral micro vessels and may manifest later in the disease’s progression [53]. Furthermore, Aβ-38 may accumulate in vessel walls as a result of alterations in the Aβ coding region of the APP, particularly in cases of severe cerebral amyloid angiopathy (CAA) [54].

#### 2.4.2. Presenilin (PSEN)

PSEN genes are the autosomal dominant form of early-onset Alzheimer’s disease (EO-AD) present in chromosome 14. The most prevalent form of familial Alzheimer’s disease (FAD) is connected to alterations in the PSEN genes, and it is also hypothesised that these mutations, though more indirectly, cause the elevated deposition of Aβ [55]. The ER membrane has nine transmembrane domains that make up the full length of PSEN [56]. It is of two types, PSEN-1 and PSEN-2.

PSEN-1 is a key protein that plays a crucial part in the synthesis of Aβ from APP by activating the α, β, and γ-secretase complex. PSEN-1 has a crucial role in maintaining memory and neurons, as demonstrated by the synaptic dysfunction and memory impairment that result from PSEN1 knockout experiments in mice [57]. By lowering the levels of Aβ-40, and alterations in the PSEN-1 gene, the ratio of Aβ-42 to Aβ-40 is raised.

PSEN-2 mutations, in contrast, are uncommon and have a negligible effect on the production of Aβ. If there are normal PSEN-1 alleles present, any mutation in PSEN-2 may significantly affect the Aβ 42/40 ratio and cause familial AD [57].

#### 2.4.3. Apolipoprotein E (Apo E)

Glycoprotein ApoE is abundantly expressed in some microglia, the liver, and the brain. It assists as a ligand for receptor-mediated endocytosis for cholesterol-containing lipoprotein particles, which are necessary for the synthesis of myelin and healthy brain working. Due to single-nucleotide polymorphisms (SNPs) that alter the coding sequence, the ApoE gene present on chromosome 19 has three isoforms, ApoE2, ApoE3, and ApoE4 [58,59]. The frequency of ApoE4 in people with AD is 2–3 times higher than in people with normal cognition. Additionally, Apo E4 is frequently linked to an early onset of the disease since it can accelerate the emergence of AD in the aged brain. A recognised indicator of AD is cerebral amyloid angiopathy (CAA) which is caused by ApoE4 activity and plays a crucial part in the deposition of Aβ as a senile plaque [60].

### 2.5. Mitochondrial Dysfunction

Mitochondria are the main source of oxidative stress due to the unavoidable leakage of electrons that occurs during electron transfer, which results in the continuous production of superoxide anion, which accounts for 90% of the endogenic ROS despite the presence of an effective mitochondrial–cellular antioxidant system [61]. It is hypothesised that defective mitochondria are less effective in producing ATP but more effective at producing ROS, which may be a significant contributor to oxidative imbalance, as seen in AD [61,62]. Certainly, a significant and early sign of AD is mitochondrial dysfunction [63]. Some of the impairments reported in mitochondria during AD are discussed below.

#### 2.5.1. Reduction in Energy Metabolism

One of the most well-known anomalies in AD is a decreased rate of energy metabolism in the diseased brain. Low glucose metabolism is actually thought to be a sensitive measure, helpful for tracking changes in cognition and functionality in AD and mild cognitive impairment (MCI). This method assists in analysis and facilitates the anticipation of future cognitive decline [64,65,66].

#### 2.5.2. Changes in the Primary Oxidative Phosphorylation Enzymes

According to a genome-wide transcriptome investigation, reduced neural expression of nuclear genes encoding components of the mitochondrial ETC may be related to cerebral glucose metabolic impairment in AD. In fact, several important oxidative metabolism enzymes, such as the pyruvate dehydrogenase complex, α-ketoglutarate dehydrogenase complex, and cytochrome oxidase, all show decreased expression in AD [67].

#### 2.5.3. Dyshomeostasis of Calcium

ER calcium channels are directly impacted by dysfunctional mitochondria, which results in decreased buffering capacity, which leads to calcium dyshomeostasis [68]. The ER develops calcium overload and calcium uptake is reduced as a result of calcium mismanagement in peripheral cells from AD patients.

#### 2.5.4. Mitochondrial DNA (mtDNA)

The brains of AD patients show elevated levels of spontaneous mtDNA alterations, including the most prevalent 5-kb deletion [69]. The mtDNA regulatory areas had a number of alterations that were specific to AD. MtDNA is susceptible to ROS attack since it lacks protective proteins such as histone, has a relatively ineffective DNA repair machinery, and is located near the place where ROS are produced. In the mtDNA of AD brains, the presence of an oxidised nucleoside indicated a three-fold increase in oxidative damage [70].

### 2.6. Vascular Factor

It was proposed by de la Torre and Mussivand in 1993. These scientists noticed that cerebral blood flow, glucose metabolism, and oxygen consumption decreased in accordance with the severity of the disease in AD patients. Several studies confirm the relation between various vascular factors and an elevated chance of developing AD. According to research, arterial hypertension raised the risk of AD in older people who had never taken antihypertensive medication [71]. Several researchers showed that middle-aged individuals with systolic hypertension and high blood cholesterol levels had a higher risk of AD in the future [72,73]. Atherosclerosis and AD are related as atherosclerotic individuals have a threefold increased risk of AD. The blood–brain barrier (BBB) plays a significant role in controlling how circulating metabolites reach brain tissue and disruption of the BBB may significantly affect the accumulation of hazardous chemicals in the brain [74]. There has been evidence that AD causes a decrease in the levels of numerous tight-junction and adherens-junction proteins and their adaptor molecules, causing changes in BBB permeability. There are pieces of evidence that the BBB plays a role in controlling Aβ levels in the CNS. RAGE, or the receptor for advanced glycation end products, controls the spread of toxicity and their transfer to the brain. RAGE expression in the brain’s endothelium offers a mechanism for the influx of Aβ and monocytes carrying Aβ across the BBB [75].

### 2.7. Immune System Dysfunction

There are numerous pieces of evidence that indicate AD patients have immune system abnormalities. Immunoglobulins, helper and cytotoxic/suppressor T-cells, circulating immune complexes (CIC) in peripheral tissue, and cerebral blood vessels complement proteins linked to the ‘classical pathway’, brain reactive antibodies, immunoglobulins, abundant reactive microglia, and more, have been seen in AD [76,77]. Complement system proteins serve as pattern recognition molecules that help glial cells that carry complement receptors that take up Aβ. Major histocompatibility locus (MHC) antigens have also been shown to undergo significant changes indicating either enhanced antigen resistance or susceptibility in AD. These immune responses may be a response to AD’s pathogenic processes, particularly the Aβ deposition [78].

A further immunological contributor to AD includes the conversion of arginine to citrulline through the activity of peptidyl-arginine deiminases (PAD), enzymes that catalyse the post-translational, and this process is known as the ‘citrullination process’. Citrullinated proteins are expressed in the hippocampus and cerebral cortex as a result of the selective expression of PAD2 and PAD4 in astrocytes and neurons, respectively. As a result, the loss of neurons, and the loss of cellular components, such as citrullinated proteins in AD, may cause an autoimmune reaction and the formation of autoantibodies [79].

### 2.8. Infections

Some studies revealed that infection might be a factor in AD. A viral invasion could activate microglia and pericytes, resulting in the development of amyloid. In addition, antibodies against the herpes simplex virus (HSV) might be present in the cerebral spinal fluid (CSF) of AD patients. HSV can cause aberrant protein production, which can lead to the paired helical filament (PHF) and neurofibrillary tangles (NFT) [80,81]. The BBB that selectively regulates the movement of molecules in and out of the brain shields the CNS with microvascular endothelial cells (pericytes and astrocytes). A wide range of microorganisms, however, can enter the BBB and cause a number of serious disorders. Viruses can directly infect endothelial cells, pass through the BBB, and enter the central nervous system, though bacteria are capable of traversing the BBB using a variety of methods, including transcellular traversal, paracellular traversal, and trojan horse, so an acute inflammatory state becomes a chronic one [82]. Uncertainty exists regarding the potential for COVID-19 to either initiate or accelerate the new onset of Alzheimer’s. One latest study demonstrating an elevated risk for SARS-CoV-2 infections in fully immunized individuals with Alzheimer’s disease was conducted using the TriNetX Analytics network technology [83]. The study’s sample included 6,245,282 older persons (age 65) who had medical interactions with healthcare organizations between 2 February 2020 and 30 May 2021 but had not previously been diagnosed with Alzheimer’s. The population was divided into COVID-19 cohorts and non-COVID-19 cohorts. Using closest neighbour greedy matching, cohorts were propensity-score matched (1:1) for demographics, negative socioeconomic health determinants, including issues with education, occupational exposure, physical, social, and psychosocial environment, and recognized risk factors for Alzheimer’s disease [84]. Within 360 days of the COVID-19 diagnosis, the likelihood of a new diagnosis of Alzheimer’s disease was estimated using a Kaplan–Meier analysis. Before propensity-score matching, the COVID-19 cohort had a total risk of 0.68% for receiving a new diagnosis of Alzheimer’s disease compared to a non-COVID-19 cohort’s 0.35%. After matching based on propensity scores, the COVID-19 cohort had a higher chance of receiving a new diagnosis of Alzheimer’s disease than the matched non-COVID-19 group [85].

## 3. Treatment of AD

There are already more than 55 million cases of AD documented globally, and by 2050, the overall number of AD patients is expected to more than triple [4,5]. Even though it is a serious health issue proper and complete treatment is not available, treatment strategies used today concentrate on assisting patients in managing behavioural symptoms, sustaining mental function, and delaying or preventing the signs of illness. Two treatment strategies can be adopted as discussed below.

### 3.1. Chemical-Based Treatment

Despite the fact that AD is a public health problem, there are currently only two classes of medications that have been approved by the FDA to treat AD: cholinesterase enzyme inhibitors (naturally occurring, synthetic, and hybrid analogues), and antagonists to N-methyl D-aspartate (NMDA).

#### 3.1.1. Cholinesterase Inhibitors

According to the cholinergic theory, a reduction in the synthesis of acetylcholine (ACh) causes AD. A reduction in acetylcholinesterase along with an increase in cholinergic levels is one therapy that enhances neuronal cell and cognitive function [86]. Acetylcholine breakdown in synapses is prevented by acetylcholinesterase inhibitors (AChEIs), leading to continuous ACh build up and cholinergic receptor activation. Another approach to treating AD may involve raising choline reuptake and, consequently, the generation of acetylcholine at presynaptic terminals. This might be done by focusing on the choline transporter (CHT1), which is in charge of supplying the choline required for the synthesis of ACh [86,87]. Different AChEIs are donepezil, rivastigmine, and galantamine.

##### Donepezil

The most effective medication for treating AD is donepezil, which is a derivative of indanonebenzylpiperidine and a member of the second generation of acetylcholinesterase inhibitors (AChEIs). Due to donepezil’s reversible binding to acetylcholinesterase, there is more ACh present at the synapses and prevents it from being hydrolysed. With transient cholinergic side effects that affect the neurological, as well as gastrointestinal systems, the medicine may be tolerated by the patient. Notably, donepezil is used to treat AD symptoms, such as improving cognition and behaviour [88,89]. Due to an imbalance in acetylcholine, unusual adverse reactions such as extrapyramidal side effects are more likely to occur when AD medication is used along with psychiatric medicines. A case of an extrapyramidal adverse response brought on by the donepezil and risperidone combination was reported [90]. The patient experienced fatigue, nausea, panic, sweating, and vomiting.

##### Rivastigmine

It is a butyrylcholinesterase (BuChE) and acetylcholinesterase (AChE) pseudo-irreversible inhibitor. In order to function, it binds to the two active sites of AChE which are estearic and anionic sites, which stops acetylcholine (Ach) metabolism [91]. In the healthy brain, glial cells contain BuChE and have only a 10% activity level compared to the AD brain, where it has a 40–90% activity level, while simultaneously reducing ACh activity. This implies that BuChE activity can be a sign of mild to severe AD. Rivastigmine is metabolised by AChE and BuChE at the synapses and dissociates slower than AChE, which is why it is known as a pseudo-irreversible. The drug is used for the treatment of mild to moderate AD. It ameliorates daily activities and cognitive processes [92,93]. The most common adverse effects of rivastigmine are gastrointestinal problems such as bladder pain, painful urination, etc.

##### Galantamine (GAL)

For mild to severe AD cases, it is regarded as a conventional first-line medication. Galantamine is a dual-mode selective tertiary isoquinoline alkaloid, which not only acts as a competitive inhibitor of AChE but also has the ability to allosterically bind to and activate the nicotinic acetylcholine receptors subunit. Like other AChE inhibitors, GAL has good efficacy and tolerability and can reduce behavioural symptoms and improve daily activities, cognitive performance, and mood [94,95]. For transporting the medicine only to the areas of the brain that were injured, it is linked to hydroxyapatite particles that contain ceria. To transport GAL hydrobromide, some researchers have used solid-lipid nanoparticles and nano emulsification techniques [96]. The results of these tests are promising for the safe administration of the drug. Nasal delivery of a GAL hydrobromide–chitosan combination of nanoparticles has good pharmacological potential, while the controlled release dose of the drug has been transported via the patch technique by another group. The common problems associated with this drug are gastrointestinal problems, headache, dizziness, insomnia, weight loss, loss of appetite, etc. [96,97].

#### 3.1.2. N-methyl D-aspartate Receptor (NMDAR) Antagonists

It is thought that NMDAR performs an important role in the pathophysiology of AD. Ca^2+^ influx brought on by NMDAR activation promotes signal transduction, and results in gene transcription that is required for the growth of long-term potentiation (LTP), which is essential for the establishment of synaptic neurotransmission, plasticity, and memory [98]. Excessive NMDAR activation overstimulates glutamate, the main excitatory amino acid in the CNS, which results in excitotoxicity, synaptic malfunction, neuronal cell death, and damage to cognitive abilities. Numerous NMDAR uncompetitive antagonists have been created and tested in clinical settings, though the majority of them were ineffective and had undesirable side effects [99]. The sole drug in this class that is approved for the treatment of moderate to severe AD is memantine.

##### Memantine

It is an uncompetitive, low-affinity antagonist of the glutamate receptor subtype. To treat mild to severe AD, memantine is administered alone or in combination with AChEI [95]. The drug has a low affinity and is quickly displaced from NMDAR by high quantities of glutamate. It blocks excitatory receptors without impairing regular synaptic communication, which makes it harmless, well tolerable, and avoids a long-lasting blockage. Possible adverse effects of memantine are dizziness, constipation, vomiting, hypertension, and headache [100].

### 3.2. Plant-Based Treatment

Currently available synthetic medicines are effective only for 1–4 years for mild to moderate AD. Synthetic medicine exhibits many negative side effects [101]. Scientific evidence related to the efficacy of phytochemicals in the prevention and treatment of AD has been accumulating which shows that they are safe and cost effective. Oxidative stress is one of the proven causes of AD. However, plants are reservoirs of antioxidants which can mitigate the effects of AD [102,103]. Several plants were examined for their ability to combat AD as listed in Table 1 and also shown in Figure 2. A diet high in plants has repeatedly been linked to a lower risk of AD. It is advised to consume fruits, vegetables, cereals, and nuts on a regular basis for overall health, to promote healthy ageing, and to reduce the risk of age-related disorders such as AD [104,105].

## 4. Plants with Anti-Alzheimer Properties

Different plants belonging to the families Solanaceae, Plantaginaceae, Fabaceae, Rubiaceae, Asteraceae, Ericaceae, Amaryllidaceae, Zingiberaceae, Pedaliaceae, Hypericaceae, Piperaceae, Lilliaceae, Ginkgoaceae, Apiaceae, Araliaceae, Polygalaceae, Crassulaceae, Lamiaceae, Apocynaceae, Theaceae, Vitaceae, Cannabaceae, Oleaceae, Lycopodiaceae, Punicaceae, Iridaceae, Lamiaceae, Caryocaracea, Arecaceae, Aloaceae, Rutaceae, Moringaceae, Juglandaceae, Lauraceae, Phyllanthaceae, Moraceae, Convolvulaceae, Halymeniaceae, Rosaceae, etc., have anti-Alzheimer properties and have been used for the treatment of AD.

### 4.1. Ginseng

*Panax ginseng* (family: Araliaceae), commonly known as ‘ginseng’ is one of the well-known herbs in China, Japan, and Korea used to treat AD. It consists of phytochemicals such as ginsenosides (saponins), a derivative of the triterpenoid dammarane, and 20(S)-protopanaxadiol, which prevents β-amyloid from aggregating and clears it from neurons, relieves mitochondrial dysfunction, and boosts the secretion of the neurotrophic factor [127,128]. According to a molecular enzyme study, ginsenosides have substantial AChE inhibitory activities, which is an efficient strategy for lowering the symptoms of AD [176,177]. Through the stimulation of phosphatidic acid receptors involved in hemolysis, the bioactive glycoprotein gintonin lowers the production of Aβ and enhances learning and memory. Additionally, it reduces AD symptoms by promoting autophagy, anti-inflammatory mechanisms, antiapoptosis, and management of oxidative stress, as proven by comprehensive in vivo and in vitro investigations [178]. Gintonin modulates the G protein-coupled lysophosphatidic acid receptors which affect the cholinergic system and neurotrophic factors, reducing the level of plaque formation. In a clinical experiment with a limited sample size of 10 people who had mild cognitive impairment or early dementia, gintonin intake (300 mg/day, 12 weeks) significantly enhanced Korean mini mental state test scores at 4 and 8 weeks compared to baseline scores. In contrast, gintonin consumption (300 mg/day, 4 weeks) significantly raised the ADAS-Cog-K and ADAS-non-Cog-K scores on the Korean cognitive subscale of the Alzheimer’s disease assessment scale after 4 weeks compared to the baseline scores. When it comes to gintonin toxicity in humans, none of the patients reported any negative side effects during the 12-week dose of gintonin. Hence, gintonin administration to older subjects with cognitive impairment was safe and well tolerated [179].

### 4.2. Gotu Kola

*Centella asiatica* (family: Apiaceae) is commonly called ‘gotu kola’. It is a widespread persistent herbaceous climber in Asia. It is used in traditional medicines for the purpose of regenerating brain cells and enhancing memory, lifespan, and intellect [134]. Animal studies have shown that *Centella asiatica* has an impact on neuronal structure, learning ability, and memory-retaining ability. It has been shown to improve cognitive performance by reducing phospholipase A2 (PLA2) activity, suppressing acetylcholinesterase activity, preventing the formation of amyloid, and preventing brain damage [180,181]. In preclinical studies, *Centella asiatica* was also discovered to have antidepressant, anxiolytic, antistress, and seizure-prevention properties [182,183]. It has been shown to affect metabolic pathways connected to AD when administered to 5xFAD mice [184]. In rats overexpressed with β-amyloid, *Centella asiatica* extract has been demonstrated to enhance memory and decision-making, while it lowers hippocampus mitochondrial dysfunctioning. In a clinical investigation, a 70% water-ethanol extract of *C. asiatica* demonstrated promising anxiolytic properties by reducing anxiety and stress in patients [184].

### 4.3. Ginkgo

*Ginkgo biloba* (family: Ginkgoaceae) is commonly known as ‘ginkgo’. It is the most well-known herb for treating Alzheimer’s and its symptoms. Terpene lactones and flavone glycosides are both present in plant extracts. The terpene lactones include bilobalide A, B, and C, and ginkgolides, while the flavone glycosides include kaempferol, quercetin, and isorhamnetin [121]. Through the control of glutathione peroxidase, catalase, and superoxide dismutase (SOD) activity, this herbal extract shields against Aβ generated neurotoxicity by preventing apoptosis of neurons, reactive oxygen species (ROS) collection, glucose assimilation, mitochondrial dysfunctioning, and activation of the extracellular signal-regulated kinase (ERK) pathway [125,126]. Numerous studies have connected astrocytosis, microgliosis, and the presence of proinflammatory substances to the deposition of Aβ peptides [185]. *G. biloba* extracts demonstrated therapeutic advantage in AD, compared to donepezil, with few unfavourable side effects. It is most recognized for its capacity to improve circulation (vasorelaxing effect) throughout the body. *G. biloba* can thus reduce blood pressure and prevent platelet aggregation [186]. In an experiment involving 18 randomized clinical trials (RCTs) with 1642 individuals, 842 of them were in the experimental group (donepezil hydrochloride plus *G. biloba* formulations) and 800 were in the control group (donepezil), it was observed that donepezil with *G. biloba* can enhance clinical efficacy rates and verbal memory. However, to validate this, more stringent trials will be required in the future [187].

### 4.4. Turmeric

*Curcuma longa* (family: Zingiberaceae) is commonly known as ‘turmeric’. Curcuminoids, such as curcumin, demethoxycurcumin, and bis-demethoxycurcumin, are the phytochemicals present in turmeric. The primary curcuminoid is curcumin, which gives turmeric roots their characteristically yellow colour. According to research, curcumin may be a potential drug for treating AD [188]. The level of oxidative damage in the brain can be reduced by curcumin. It has been shown that curcumin can reverse β-amyloid pathology in a mouse model with AD [189]. The antioxidant and anti-inflammatory properties of curcumin also facilitated in alleviating of some AD symptoms [118,119]. The capacity of the Early Growth Response-1 (Egr1) protein to bind DNA is inhibited by curcumin, which reduces inflammation. Activated microglia and astrocytes produce chemokines which are known to cause monocyte chemotaxis and are also inhibited by curcumin at the CNS. Effective ways to stop proinflammatory cytokine activation include decreasing the production of ROS by stimulating neutrophils and suppressing the tumor necrosis factor α (TNF-α) and interleukin-1 (IL-1) inflammatory cytokine expression [190,191]. Curcumin inhibits the activity of the activator protein (AP-1), a transcription factor involved in the synthesis of amyloid. The capacity of curcuminoids to prevent the generation and spread of free radicals is proof that they possess potent antioxidant effects. It also prevents the oxidation of free radicals and low-density lipoproteins which causes the destruction of neurons in AD and other neurodegenerative diseases.

### 4.5. Brahmi

*Bacopa monnieri* (family: Plantaginaceae) commonly known as ‘brahmi’ is a persistent creeper that is indigenous to the swamps of eastern and southern India, together with Australia, Europe, Africa, Asia, North and South America, and the Middle East. In traditional medicine, it is frequently used as a cardiotonic, diuretic, and nerve tonic [192,193]. The main phytochemicals of Brahmi are Brahmine, bacosides A and B, apigenin, quercetin, bacosaponins A, and bacosaponins B. Protein kinase activity is increased by *B. monnieri* extracts, which has a nootropic effect. Rats administered Brahmi extract displayed reduced cholinergic degradation and an improvement in cognition. Additionally, it also shields neural cells from the harm done by β-amyloids [193]. *B. monnieri* extract treatment resulted in decreased ROS levels in neural cells, indicating that it reduces intracellular oxidative stress. Cognitive abilities significantly increase with regular use of Brahmi, which also reduced their levels of inflammation and oxidative stress [194]. In addition, a team of researchers found that an extract of standardised *B. monnieri* corrected the cognitive abnormalities brought on by the intracerebroventricular administration of colchicines and ibotenic acid into the nucleus basalis magnocellularis. In the same study, *Bacopa monnieri* also restored acetylcholine depletion, choline acetyltransferase activity reduction, and reduction of muscarinic cholinergic receptor binding in the frontal cortex and hippocampal regions [195]. By suppressing cellular acetylcholinesterase activity, Brahmi extracts prevent beta-amyloid-induced cell death in neurons. In a study (randomized, double-blinded trial) involving 81 persons of the age group 55 and above, a 12-week cycle of Bacopa considerably improved memory acquisition and retention [196].

### 4.6. Ashwagandha

*Withania somnifera* (family: Solanaceae) is commonly known as ‘ashwagandha’ and is regarded as a Rasayana (rejuvenating). It possesses antioxidant properties, characteristic of free radical scavengers. The chemical composition of ashwagandha root includes alkaloids, anolides, many sitoindosides, and flavonoids [197,198]. According to a molecular study, ashwagandha root helps in treating AD by preventing nuclear factor B activation, promoting nuclear factor erythroid 2-related factor 2 (Nrf2) migration to the nucleus, where it enhances the expression of antioxidant enzymes, to reduce the formation of amyloid, decrease apoptotic cell death, restore synaptic function, and boosts the immune system [199]. In certain research, ashwagandha root methanolic preparations were used to treat human neuroblastoma SK-N-SH cells, which led to an increase in dendritic extension, neurite outgrowth, and synapse formation. Researchers have hypothesised that the ashwagandha root extracts are effective in treating neurodegenerative illnesses and also promote neurite growth, and have anti-inflammatory, antiapoptotic, and anxiolytic effects. Moreover, they have the capacity to minimise mitochondrial dysfunctioning, boost antioxidant defence levels, reduce glutathione levels, and can cross the blood–brain barrier and reduce inflammation in the brain [200]. In a double-blind, randomized, placebo-controlled study, 50 participants with moderate cognitive impairment (MCI) were treated with a 300 mg dose of *W. somnifera* root extract twice daily for an eight-week period. After eight weeks, the *W. somnifera*-treated group displayed considerable improvements in their ability to process information, concentrate, and use executive functions [201].

### 4.7. Saffron

*Crocus sativus* (family: Iridaceae) commonly known as ‘saffron’, possesses antioxidant, anticancer, and aphrodisiac properties and also improves memory in adults. Numerous studies have shown that saffron possesses antioxidative, anti-inflammatory and antiamyloidogenic properties. Additionally, saffron is said to be helpful in reducing acetylcholinesterase and protecting against toxins (AChE). AChE is connected to the neurofibrillary tangles and beta-amyloid plaques that are characteristic of AD [202].

To analyse the effect of saffron on learning abilities, and the prevention of oxidative stress, each rat was administered five and ten grams of saffron extract, twice a week. Oxidative stress markers were assessed seven days later. The group that received saffron treatment was found to have a reduced memory deficit along with enhanced spatial learning and antioxidant activity of enzymes [203]. The main bioactive compound of saffron is crocin. It has the ability to bind to the hydrophobic region of Aβ and thus inhibits its aggregation [204]. A double-blinded/phase II study using the AD assessment scale, cognitive subscale, clinical dementia rating scale, and sums of boxes scores was conducted on a total of 54 patients who were 55 years of age or older with AD. These patients received saffron extractive (30 mg) or donepezil (10 mg) as a positive control once daily for 22 weeks. As a result, donepezil and saffron extractives had similar effects on patients with mild to moderate AD, suggesting that saffron extractives have a therapeutic effect [154].

### 4.8. Ginger

*Zingiber officinale* (family: Zingiberaceae) commonly called ‘ginger’ is a spice having both culinary and therapeutic uses. It is frequently used as a nutritional supplement, in ginger tea preparation, or as an extract. The primary bioactive components in ginger include gingerols, shagols, volatile oils such as bisabolene and zingiberene, and monoterpenes. In vitro research has been done on the AChE inhibitory activity of red and white ginger [205]. Inhibition of AChE causes acetylcholine to accumulate in synapses, which is followed by an increase in the cholinergic pathway activity and results in better cognitive performance in AD patients.

Ginger’s ability to decrease lipid peroxidation is vital for the prevention of AD. Pro-oxidants such as quinolinic acid (QUIN) and sodium nitroprusside (SNP) are utilised to cause lipid peroxidation in the rat-brain homogenate. Due to the overstimulation of NMDA receptors and the significant rise in malondialdehyde level brought on by the incorporation of SNP and QUIN, free radicals are produced [155]. Ginger extract was demonstrated to boost brain SOD and CAT expression, decrease NF-ĸB, interleukin-1 beta (IL-1β), and malondialdehyde (MDA) levels and improve behavioural impairment in a rat model of AD caused by oral AlCl_3_ and injection of intracerebroventricular β-amyloid protein [206]. In a similar study, the fermented ginger extract had more bioavailability and has been shown to greatly reduce synaptic dysfunction and neuron cell loss, compared to the fresh extract, in a mouse model of AD produced by injection of β-amyloid plaques [207].

### 4.9. Rosemary

*Rosmarinus officinalis* (family: Lamiaceae) is commonly called ‘rosemary’. Other than its native Mediterranean region, several other countries are known to use the plant in traditional medicine.

It possesses antioxidant and anti-inflammatory properties. To learn how drinking rosemary tea affects the working of the brain, an investigation on adult male mice was done. The testing revealed that rosemary tea consumption for four weeks had a favourable effect (anxiolytic- and antidepressant) without changing the memory or learning [112]. Other researchers have shown that it possesses antidepressant properties and is able to reverse ACHE changes despite spatial learning impairment [208]. Carnosic acid has also been found to have neuroprotective effects on cyanide-induced brain damage in cultured rodent and human-induced pluripotent stem cell-derived neurons in vitro and in vivo in several brain locations in a non-Swiss albino mouse model [209]. In vitro, the intercellular adhesion molecule (ICAM-1) expression is suppressed and tumour necrosis factor (TNF)-induced monocyte adherence to endothelial cells is inhibited by carnosol and rosemary essential oils [210]. Carnosol decreases the activity of the nuclear factor kappa-B inhibitor and increases the production of heme oxygenase-1 (HO-1), both of which block the signalling pathways triggered by TNF-α [211]. According to a study conducted on 68 students in Kerman, Iran, using 500 mg of rosemary twice daily for a month improved students’ prospective and retrospective memory [212].

### 4.10. Date Palm

*Phoenix dactylifera* (family: Arecaceae) is commonly called ‘date palm’. They have been used since Mesopotamian civilization, and their historical, theological, and medicinal significance is well known [213,214]. Three to four date fruits per day were recommended for improving memory in Palestine [214]. Turkish people drink “Hurma coffee,” an herbal brew made from date fruit seeds, to improve their memory. It reduces glutathione, glutathione reductase, and glutathione peroxidase levels [215]. In addition, mice with AD were fed diets supplemented with 2 and 4% acetone-extracted date fruit, for 14 months, and the results were compared to mice receiving a control diet. When mice were fed dates at 2 and 4% levels, oxidative stress markers such as protein carbonyl levels, lipid peroxidation, and the restoration of anti-oxidative stress enzymes were all considerably reduced [216].

### 4.11. Pumpkin Seeds

*Cucurbita maxima* (family: Cucurbitaceae) is commonly known as ‘pumpkin’. Pumpkin seeds are included in the category of nuts. Despite their significant nutritional content and therapeutic qualities, pumpkin seeds are typically seen as agricultural waste and are thrown away. In addition to being added to food preparations, they can be eaten in their fresh or roasted form. Pumpkin seeds are rich in choline (63 mg/100 g) and L-tryptophan (576 mg/100 g) [94]. L-tryptophan is frequently used to treat a variety of medical disorders, including anxiety, sleeplessness, and depression [217,218]. The body can convert tryptophan to serotonin, which in turn may control a number of cognitive functions. It is known that choline serves as a precursor for the synthesis of the neurotransmitter acetylcholine in cholinergic synapses, which deliver stimulatory signals to nerve cells. Moreover, choline promotes brain growth [219]. In adult male Wistar rats, oral treatment of pumpkin-seed oil (100 mg/kg and 200 mg/kg for 5 days) is reported to have antiamnesic benefits against scopolamine-induced amnesia. It suppresses acetylcholine esterase, reduces TNF expression in the hippocampus, and raises glutathione levels in the brain [219].

### 4.12. Garlic

*Allium sativum* (family Liliaceae) is commonly known as ‘garlic’. It is widely used in traditional medicines for the treatment of numerous diseases, including AD. The most popular garlic preparation used is called AGE, and it is often made by keeping slices of garlic in a solution of water and ethanol for more than 10 months at ambient temperature. Aggregation of unusually folded Aβ and tau proteins in amyloid plaques and neuronal tangles are the main pathologies of AD. The two primary types of Aβ are Aβ40 and Aβ42. AGE at dosages of 250 and 500 mg/kg BW can improve short-term memory deficits in humans [123,124]

It has been discovered that raw garlic has strong antineuroinflammatory capabilities, and this is due to organosulfur compounds (OSCs) that are produced from alliin (such as allicin, diallyl trisulfide, and diallyl disulfide). In lipopolysaccharides (LPS)-activated microglial cells, these substances, particularly diallyl trisulfide and diallyl disulfide, reduce the generation of TNF-α, lipopolysaccharide (LPS) induced nitric oxide, monocyte chemoattractant protein-1, and interleukin-1 (IL-1) [220]. Similar to this, glial cell activation caused by LPS and inflammatory mediators that are implicated in amyloidogenesis is reduced by the sulphur-containing substance thiacremonone [221].

## 5. Phytochemicals

Phytochemicals have long been employed as treatment options for a number of pathological conditions, and a balanced diet rich in phytochemicals can reduce the risk of AD [107]. The mechanisms of many phytochemicals have been discussed and, for some phytochemicals, it has to be established yet, and their amount in food that makes them bioavailable is still under research [116]. Phytochemicals have been shown in in vitro and in vivo investigations to have a possibility for AD treatment, allowing for a few of them to go into the clinical trial phases [187]. According to research, phytochemicals can raise α-secretase activity, decrease Aβ formation, reduce tau hyperphosphorylation, increase antioxidant enzymes, and improve learning and memory [185,190,200], and shows significant potential in treating AD by acting on various mechanisms, as shown in Figure 3.

### 5.1. Huperzine A

A substance called Huperzine A was produced from a specific kind of club moss (*Huperzia serrata*). *H. serrate* extract can be utilized as a dietary supplement to enhance memory. Huperzine A has a significant impact on AChE inhibition. Its mechanism is comparable to that of the anti-AD drugs galantamine, donepezil, and rivastigmine [222]. According to clinical studies, huperzine A has extremely few negative side effects, such as stomachaches and headaches. Huperzine A also decreases oligomeric and β-amyloid plaques in the cortex and hippocampus, respectively. Additionally, huperzine A can block the brain’s NMDA receptor and potassium channel [223,224].

### 5.2. Epigallocatechin-3-gallate

A catechin of the flavonoid group called epigallocatechin-3-gallate is found in *Camellia sinensis*. Numerous researchers have examined the impact of epigallocatechin-3-gallate on a wide range of illnesses, including cancer and cardiovascular and neurological disorders [225,226]. Strong antioxidant activity is exhibited by epigallocatechin-3-gallate. In mice with streptozotocin-induced dementia, epigallocatechin-3-gallate has been demonstrated to boost glutathione peroxidase activity, reduce AChE activity, and prevent the accumulation of NO metabolites and ROS [227]. In mutant PS2 Alzheimer mice, epigallocatechin-3-gallate also improved memory and reduced the activity of the enzyme γ-secretase. Epigallocatechin-3-gallate also reduced amyloid precursor protein expression, decreased the activity of enzyme one that cleaves beta-sites from APP, and decreased β-amyloid buildup to defend against apoptosis and memory loss brought on by LPS [141].

### 5.3. Resveratrol

Resveratrol is a polyphenolic substance that is a member of the stilbene family. Almonds, grapes, and other fruits contain resveratrol. Numerous research has demonstrated that it possesses cardiovascular, anticancer, anti-inflammatory, antioxidant, and blood-glucose-lowering characteristics, as well as a neuroprotective impact. By scavenging ROS, raising glutathione levels, and enhancing endogenous antioxidants, resveratrol exerts a powerful antioxidant effect [228]. By triggering APP’s nonamyloidogenic cleavage and enhancing β-amyloid clearance, resveratrol can also lower levels of β-amyloids. Additionally, resveratrol reduced AChE activity in neural cells. Resveratrol was shown to be safe, well tolerated, and to be able to reduce cerebrospinal fluid (CSF) and plasma A40 levels in AD [229]. 

### 5.4. Rosmarinic Acid

Many Lamiaceae plants contain polyphenol-type carboxylic acid and rosmarinic acid. Rosmarinic acid is associated with antioxidant, antibacterial, anti-inflammatory, anticancer, antiviral, and neuroprotective properties [230]. The ability of rosmarinic acid to decrease NF-ĸB and TNF expressions is the mechanism by which it greatly reduces amyloid-induced memory loss [231,232]. Additionally, it has been demonstrated that rosmarinic acid shields neuronal PC12 cells against cytotoxicity brought on by beta-amyloid. It could lessen the tau protein’s hyperphosphorylation. According to a rat model of AD, by lowering lipid peroxidation and inflammatory processes, rosmarinic acid assists in lessening locomotor activity, short-term spatial memory, and metabolic changes in brain tissue [232].

### 5.5. Galantamine

Galantamine has been utilized for age-related cognition or memory. This selective, reversible, and competitive inhibitor of AChE was first obtained from snowdrops and is presently commercialized for preventing neurological deterioration and in the treatment of AD. Galantamine is also extracted from the Narcissus species [117]. In the 1950s, a Bulgarian pharmacologist observed individuals using the common snowdrop growing in the wild and applying it on their skin to relieve the discomfort of their foreheads [233]. However, the first study to demonstrate the acetyl cholinesterase inhibitory activities of galantamine isolated from Galanthus was reported by Mashkovsky and Kruglikova-Lvov in 1951.

### 5.6. Curcumin

Curcumin, a key chemical component of turmeric (*Curcuma longa*), is used as a spice to provide taste and colour to Indian curries, as well as for preserving food. It is interesting to note that compared to the United States, AD prevalence among adults aged 71 to 80 is 4.4 times lesser in India [234,235]. There is strong in vitro evidence that curcumin possesses anti-inflammatory, antioxidant, and antiamyloid properties. Curcumin prevents lipid peroxidation, stimulates glutathione S-transferase, and increases heme oxygenase-1 (HO-1). Due to its potent inhibition of COX-2 and lipoxygenase, curcumin has been demonstrated to have anti-inflammatory properties. Additionally, curcumin inhibits iNOS and is a powerful inhibitor of NF-ĸB and AP-1 initiation. Important phases in the pathophysiology of AD include the accumulation of Aβ into fibrils and the subsequent development of amyloid plaques. Curcumin has been discovered to destabilize preformed Aβ fibrils and limit Aβ fibril production and extension in a dose-dependent manner between 0.1 and 1 M [236]. According to a clinical trial on AD mice, those given low doses of curcumin had a 40% lower level of beta-amyloid than those who weren’t given curcumin [237]. The health advantages of 80 mg/day of lipidated curcumin were investigated in a four-week clinical experiment. According to the study, plasma levels of Aβ (1–40) were reduced [238]. Sine AD is a multifactorial disorder involving many pathological mechanisms. Treatments focusing on a single causative or modifying factor will likely have limited advantages. As a result, there is increased interest in therapeutic drugs such as curcumin with a pleiotropic effect that targets numerous pathological mechanisms [239]. Cox et al. (2015) demonstrated that supplementation with solid lipid curcumin formulation (80 mg as Longvida^®^) increased cognitive function and decreased fatigue and psychological stress in an older population, suggesting curcumin has protective properties against neurodegeneration [240].

### 5.7. Caffeic Acid

Caffeic acid is abundantly present in coffee, tea, and wine, and shows a variety of pharmacological effects, including immunological modulation, antioxidant, and anti-inflammatory effects. Recent research has shown that caffeic acid has the ability to protect against toxicities caused by acrolein, 6-hydroxydopamine (6-OHDA), 1-methyl-4-phenyl-1,2,3,6-tetrahydropyridine (MPTP), and 6-hydroxydopamine (6-OHDA), as well as oxidative brain damage caused by hydrogen peroxide and stroke. It has been demonstrated that caffeine protects PC12 cells from amyloid-β-induced neurotoxicity [110]. Additionally, caffeic acid has been shown to increase acetylcholine levels in the brain and promote learning and memory. Tau proteins and β-amyloid peptides are the major constituents of plaques that are detected in the brains of people with AD. Tau phosphorylation is a crucial stage in the formation of tau protein aggregates and may potentially be involved in the onset of amyloid toxicity. Glycogen synthase kinase-3 beta (GSK3) is one of the kinases that phosphorylate tau protein; insulin signalling decreases the activity of this kinase. Thus, it is hypothesised that GSK3 dysregulation in neurons may be a key factor in the onset of AD [154]. Caffeic acid restored glutathione levels and glycogen synthase kinase 3 (GSK3) activity in the brains of hyperinsulinemic rats, inhibited GSK3 activity, and lower levels of β-amyloid and phosphorylated tau protein [241]. NF-B-p65 protein expression, oxidative stress, inflammation, and caspase-3 activity are all reduced by the intake of caffeine.

### 5.8. Silymarin

Silymarin is a combination of flavonolignans, flavonoids, and other polyphenolic chemicals that are derived from milk thistle (*Silybum marianum*), a perennial or biennial plant (family: Asteraceae) that is commonly grown in the Mediterranean region [242,243]. The anti-injury and memory-impairing properties of silymarin make it a valuable tool. Animal models of neurodegenerative illnesses, as well as neuronal and non-neuronal cellular models, have provided evidence for silymarin’s neuroprotective effects [111]. Additionally, the capacity to halt the course of neurodegeneration was examined in the AD model *Caenorhabditis elegans* CL4176. Chronic silymarin therapy for APP transgenic mice alleviated AD-like symptoms, decreased cerebral plaque and brain microglial activation, and improved the behavioural abnormalities brought on by AD disease. Silymarin dramatically increased cell survival and reduced behavioural abnormalities in APP-transgenic mice by preventing the Aβ fibrilization and deposition that occurs when APP is overexpressed in the brain [244]. It can greatly reduce the high level of TNF-α and increase the percentage of NF-ĸB mRNA expression brought on by aluminium in the rat cerebral cortex and reduce the memory deficit [242].

## 6. Algal Phytochemicals for Prevention of AD

Micro- and macroalgae are eukaryotic photosynthetic organisms that are found in tropical and intertidal environments. They are classified into different types, such as red algae, green algae, brown algae, diatoms, dinoflagellates, etc., having different shapes and sizes [245,246]. To date, approximately 32,000 compounds with a wide range of applications have been discovered [247]. Compounds having anti-Alzheimer’s properties have been discussed.

### 6.1. Fucoidan

Fucoidan is a sulphated polysaccharide obtained from brown algae. According to certain reports, fucoidan has an impact on the inflammation process at various stages. It inhibits several enzymes, prevents lymphocyte adhesion and invasion, and triggers apoptosis [248]. As caspase-9 and caspase-3 play a significant role in apoptosis processes, the ability of fucoidan to limit their activation raises the possibility that fucoidan primarily prevents neuronal death by inhibiting apoptosis. It has been reported that fucoidan therapy can lessen the repressive effects of amyloid-beta on protein kinase C (PKC) phosphorylation [249,250]. Some studies reveal that fucoidan decreases the production of ROS and TNF-α in lipopolysaccharide (LPS)-induced primary microglia [251].

### 6.2. Phlorotannins

Phlorotannins are polyphenols extracted from the brown algae species *Ecklonia stolonifera*, *Ecklonia cava*, and *Eisenia bicyclis*. The significant neurotransmitter in the brain, acetylcholine, is increased by phlorotannins such as phlorogucinol, eckol, dieckol, phloroeckol, and phlorofurofucoeckol by decreasing the action of the enzyme acetylcholinesterase and butyrylcholinesterase. Hence, the discovery that phlorotannin inhibits the BACE-1 enzyme shall enhance the AD treatment regime. It was recently demonstrated, for the first time, that the phlorotannin dieckol controls the PI3K/Akt/GSK-3β signalling pathway, which in turn controls APP proteolytic processing and Aβ synthesis [252].

### 6.3. Homotaurine

Homotaurine is a tiny natural amino sulfonate molecule that was initially isolated from several types of marine red algae. It was later chemically synthesised and utilised in medicine as tramiprosate [253]. In three phase II and three phase III clinical investigations, homotaurine’s therapeutic effectiveness in treating AD was examined. Due to its unique antiamyloid action and affinity for type A-aminobutyric acid receptors, it also offers a pertinent neuroprotective effect [254,255]. According to a therapeutic mechanism, tramiprosate is an antineurotoxic drug that inhibits the synthesis of neurotoxic amyloid-oligomers by coating the amyloid peptide to stop it from misfolding.

### 6.4. Spirolides

Spirolides are a new class of lipophilic marine toxins produced by the dinoflagellates *Alexandrium ostenfeldii* and *Alexandrium peruviaunum* [256]. They interact with neuronal nicotinic acetylcholine (nAChR) receptors and muscle types to exert their effect. No human toxicity has been documented. The leading member of this category is 13-desmethyl spirolide C, and it resulted in elevated levels of N-acetyl aspartate (NAA), which had healing effects on AD symptoms; 13-Desmethyl spirolide C has anti-AD properties and can penetrate the blood-brain barrier [257].

Many other compounds from algae such as caulerpin, racemosin A-C, caulersin, fucosterol, fucoxanthin, and α-Bisabolol have the potential to attenuate the symptoms of AD as they are reported to show anti-inflammatory and anticholinesterase activities.

## 7. Future Prospective

Alzheimer’s disease is a complex illness brought on by a series of accumulating hereditary and environmental risk factors. Finding the best treatment has proven to be particularly challenging due to the varied nature of factors contributing to the disease, as one medication will not be effective in all cases. The numerous failures during the clinical trials in the treatment of memory loss could be due to a number of factors, including a delay in initiating therapies during the course of the disease, inadequate medicine dose, wrong target for treatment, and, most significantly, little knowledge of the cause of memory loss and neurodegeneration.

Finding the root cause and creating new treatment options that address AD’s numerous pathways are urgently needed. Herbal medicines could be utilized as an alternative for neurodegenerative diseases and could also make patients feel better. Herbal drugs have been time tested and provide a variety of synergistic effects, are bioavailable, less harmful than their synthetic counterparts, and enhance cognition.

Psychoeducation, meticulous pharmaceutical, environmental and social treatment regimes, as well as dementia care are essential for the effective treatment of AD. There are numerous studies that are being conducted to evaluate the efficacy of an Alzheimer’s vaccine (having a constituent of an antigenic amyloid protein, amyloid enzyme inhibitors, and nerve growth-factor therapies). Edible vaccines can also be designed and synthesized using the techniques of genetic engineering.

The limitation of using a plant-based treatment for AD is the slow response and the requirement of having a large amount. Some phytochemicals have low bioavailability and are not absorbed by the body, and also do not reach the target site. For solving this problem, Phytotherapeutics and a nanomedicine approach (green nanotechnology) can be used for the targeted delivery of the drug. Plant-based nanoparticles (e.g., such as those synthesized from the bark of the *Terminalia arjuna*) can be used, as these nanoparticles will be less toxic than metallic ones [258]. Further studies in the field of green nanotechnology may open up new vistas for sustainability in the treatment of AD.

Homotaurine or tramiprosate is the only plant-based (algae) compound in the clinical phase. The expenses associated with bringing a novel treatment to market after its invention, clinical testing, and approval, and converting these compounds into usable pharmaceuticals are the most significant hurdles. A large amount of biomass, year-round availability, processing and marketing cost, and public acceptability are bottlenecks in the process of developing drugs from marine flora. Thus, marine-based medications may not be available in the market until the supply is managed in a way that is both commercially and ecologically viable.

These phytochemicals can also be used to fight against many viral diseases such as chikungunya, hepatitis, measles, and COVID-19 as these phytochemicals inhibit the entry of viruses into the body, destroy their genetic material and nucleocapsid, and inhibit replication. Finding the exact mode of action of different phytochemicals will help in combating this pandemic virus by creating powerful treatments and therapies.

## 8. Conclusions

Many natural substances have shown promise for treating AD in both in vitro and in vivo investigations. However, clinical trials are still required to confirm the safety and effectiveness of these substances, due to physiological variations between tested animals and human subjects. Most phytochemical clinical trials are done with a small number of participants and for a short period of time. The conflicting findings from these clinical trials imply that larger-scale studies with longer treatment periods will be necessary to validate or disprove the therapeutic efficacy of these phytochemicals in the treatment of AD. Herbal medicines are easily procurable, have several synergistic effects, including an increase in cognitive and cholinergic functioning, are bioavailable, and are substantially less toxic. They can also easily cross the blood–brain barrier (BBB). Due to the small sample sizes used in some of the clinical trials with natural substances for the treatment of AD, no definitive findings were obtained. Yet, several substances demonstrated safety in human testing and were permitted to move on to later stages. As aforementioned, herbal medications seem to be a potential and sustainable alternative therapy for AD patients. However, indepth studies on each herb in terms of extraction methodology, dosage, consortium, mode of action, efficacy, etc. in carefully planned clinical trials are required for the sustainable treatment of AD.

## Figures and Tables

**Figure 1 life-13-00999-f001:**
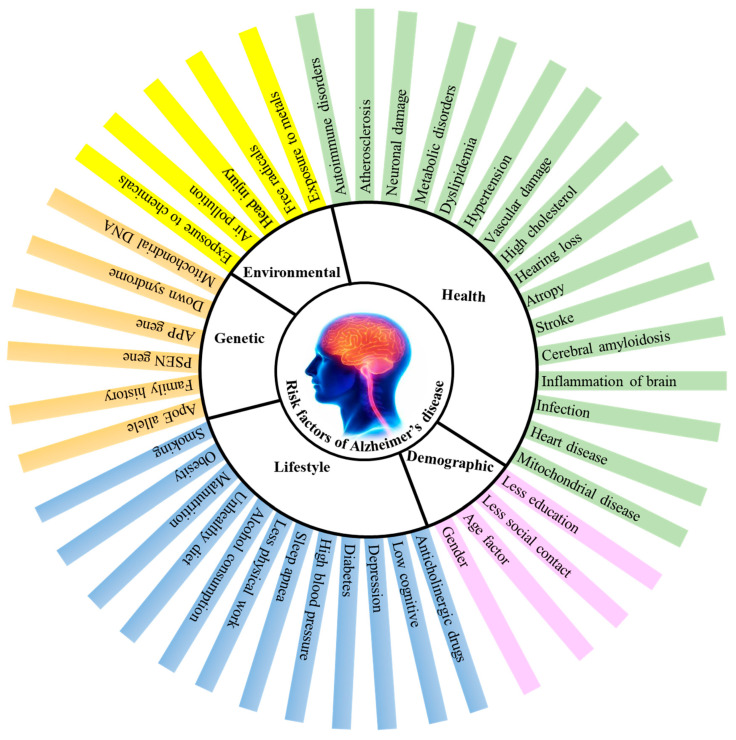
Factors contributing to Alzheimer’s disease (AD). APP-amyloid precursor protein, PSEN-presenilin, ApoE-apolipoprotein E.

**Figure 2 life-13-00999-f002:**
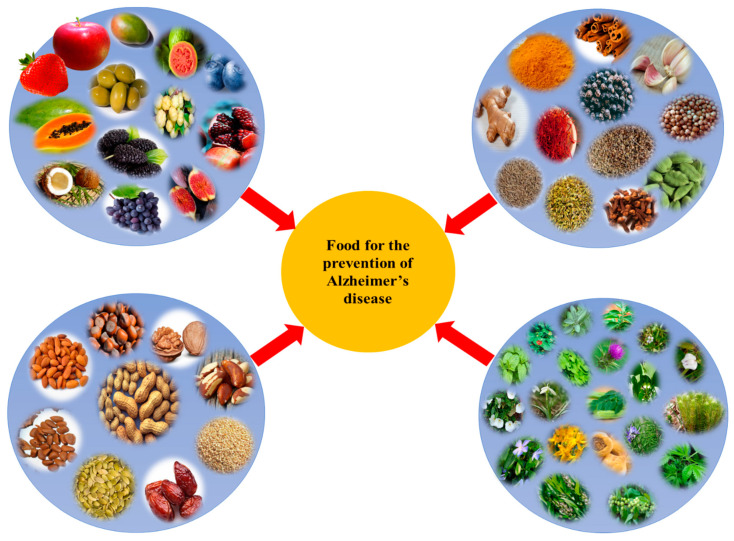
Different plant-based foods used for the prevention of Alzheimer’s disease (AD).

**Figure 3 life-13-00999-f003:**
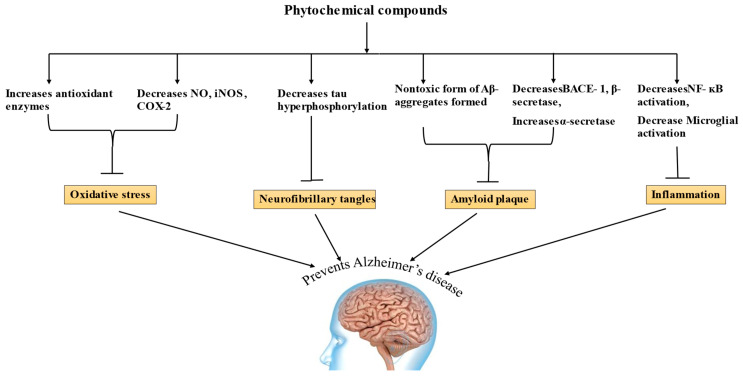
Mechanism of Alzheimer’s disease (NO—nitric oxide, iNOS—inducible nitric oxide synthase, COX-2—cyclooxygenase 2, BACE 1—Beta site amyloid precursor protein cleaving enzyme, NF-kB—nuclear factor kappa B).

**Table 1 life-13-00999-t001:** Different plants possessing anti-Alzheimer properties.

Plant	Botanical Name	Family	Part Used	Active Compounds	Properties	References
Ashwangdha	*Withania somnifera*	Solanaceae	Roots	Glycowithanolides (Withaferin A, Withasomniferin A)	It has neuroprotective functions.	[106,107]
Brahmi	*Bacopa monnieri*	Plantaginaceae	Arial parts	Brahmine, bacosides A and B, apigenin, and quercetin	It works as a memory enhancer.	[108]
Calabar bean	*Physostigma venenosum*	Fabaceae	Seeds	physostigmine	It has acetylcholinesterase inhibitor activities.	[109]
Coffee	*Coffea arabica*	Rubiaceae	Seeds	Caffeic acid, chlorogenic acid	It is effective against Alzheimer’s disease.	[110]
Milk thistle	*Silybum marianum*	Asteraceae	Seeds	Silymarin	It acts as a scavenger of free radicals and protects the central nervous system against any injury and memory impairment.	[111]
Guggulu	*Commiphora wightii*	Burseraceae	Bark	Ferulic acid, commiphoric acid, eugenol, and commophorinic acid	It acts as a scavenger of superoxide radicals.	[112]
German chamomile	*Matricaria recutita*	Asteraceae	leaves	apigenin	It helps in stimulating the brain and calms the nerves.	[113]
Blueberry	*Vaccinium corymbosum*	Ericaceae	Fruit	Antioxidants, vitamins C, B, β-carotene, lutein, and zeaxanthin	It has anti-inflammatory and antidiabetic, properties, and also helps in preventing Alzheimer’s disease.	[114,115]
Rosemary	*Rosmarinus officinalis*	Lamiaceae	Leaves	Carnosic acid, carnosol, rosemanol, rosmarinic acid, and α-pinene	It has antioxidant properties and reduces the risk of AD.	[116]
Snowdrop	*Galanthus nivalis*	Amaryllidaceae	Bulbs	Galanthamine, nivalidine, narwedine, and lycorine	It has antioxidant and antiamyloid activities.	[117]
Turmeric	*Curcuma longa*	Zingiberaceae	Rhizome	Curcumin, bisdemethoxycurcumin, eugenol demethoxycurcumin, zingiberene dihydrocurcumin, azulene, D-camphene, caprylic acid, cineol, and turmerone	It has antioxidant properties so it helps in preventing Alzheimer’s disease.	[118,119]
St. John Wort	*Hypericum perforatum*	Hypericaceae	Entire plant	quercetin, Hypericin, rutin quercetin, and isorhamnetin,	It possesses antioxidant and antiamyloid activities.	[120,121]
Black pepper	*Piper nigrum*	Piperaceae	Seeds	piperine	It reduces acetylcholinesterase levels and shows better results in the treatment of Alzheimer’s disease.	[122]
Garlic	*Allium sativum*	Lilliaceae	Cloves	S-allyl-cysteine, S-allyl-mercaptocysteine Biophenols: caffeic acid, and ferulic acid	It shows antiamyloid and antitangle properties.	[123,124]
Ginkgo	*Ginkgo biloba*	Ginkgoaceae	Leaves	Ginkgolides A, B, C, J and M, bilobalide, quercetin, sesquiterpene kaempferol, and isorhamnetin	It has antioxidant properties. It increases the blood flow in the brain and acts as a scavenger of free radicals and shows neuroprotective properties.	[125,126]
Coriander	*Coriandrum sativum*	Apiaceae	Leaves	Camphor, limonene, alpha-pinene, geraniol, petroselinic acid, and linalool	It helps in improving memory and also helps in managing Alzheimer’s disease.	[127,128]
Sesame	*Sesamum indicum*	Pedaliaceae	seeds	Sesaminol, sesamine	It shows neuroprotective properties.	[129]
Apple	*Malus pumila*	Rosaceae	Fruit	Quercetin, catechin, and epicatechin	It improves cognitive functions.	[130]
Ginseng	*Panax ginseng*	Araliaceae	Roots	Ginsenosides, gintonin	It improves the functioning of the central nervous system, and it also shows anti-amyloid activity.	[131,132]
Mulberry	*Morus alba*	Moraceae	Fruit	resveratrol, oxyresveratrol, chlorogenic acid, mulberroside, moracin, and maclurin	It has antioxidant properties and helps in lowering the risk of AD.	[133]
Gotu kola	*Centella asiatica*	Apiaceae	Leaves	Quercetin, myricetin, kaempferol, rutin, and apigenin	It possesses anti-amyloid properties.	[134]
Seneca snakeroot	*Polygala tenuifolia*	Polygalaceae	Roots	Tenuigenin, tenuifolin, and xanthone glycosides	It acts as an acetylcholinesterase and beta-secretase 1 inhibitor.	[135,136]
Golden root	*Rhodiola rosea*	Crassulaceae	Roots	Rosavin, salidroside, rosin, cinnamoyl alcohol, and tyrosol	It has very good antioxidant activity and also acts as a cognitive enhancer.	[137,138]
Lemon balm	*Melissa officinalis*	Lamiaceae	Leaves	Citral, protocatechuic acid, caffeic acid, and rosmarinic acid	It acts as a memory enhancer.	[139]
Dwarf periwinkle	*Vinca minor*	Apocynaceae	Upper parts	Vinpocetine, apovincaminic acid, kaempferol glycosides, hydroxybenzoic acids, and chlorogenic acid	It acts as a memory enhancer and also shows antioxidant properties.	[140]
Green tea	*Camellia sinensis*	Theaceae	Leaves	Gallocatechin, Gallic acid, epigallocatechin, epicatechin, epigallocatechin gallate, and caffeine	It possesses antioxidant and antiamyloid activities.	[141,142]
Grapes	*Vitis vinifera*	Vitaceae	Fruit	Resveratrol, quercetin, and catechins	It has antioxidant and antiamyloid properties and is used in preventing neurodegeneration.	[143]
Marijuana	*Cannabis sativa*	Cannabaceae	Bud and leaves	Tetrahydrocannabinol, cannabidiol	It shows antiamyloid activity.	[144]
Olive	*Olea europaea*	Oleaceae	Fruit, oil, leaves	Oleuropein, tyrosol, hydroxytyrosol, caffeic acid, verbascoside, and rutin	It possesses antioxidant, anti-inflammatory, and antiamyloid properties.	[145]
Brazil nut	*Bertholettia excelsa*	Lecythidaceae	Nut	Lecithin	It increases the level of acetylcholine n AD patients.	[146]
firmoss	*Huperzia serrata*	Lycopodiaceae	Aerial parts	Huperzines	It possesses antiamyloid activity.	[147]
Pomegranate	*Punica granatum*	Punicaceae	Fruit	Ellagic acid, gallagic acid punicalagin, and punicic acid	It possesses antioxidant and antiamyloid activities.	[148,149]
Marapuama	*Ptychopetalum olacoides*	Olacaceae	Roots	Ptychonal, muirapuamine, and theobromine	It possesses antiamnesic, anticholinesterase, and neuroprotective properties.	[150,151]
Fennel	*Foeniculum vulgare*	Apiaceae	Seed	Estragole, limonene, fenchone, and β-myrcene	It shows an inhibitory effect against acetylcholinesterase and butyrlcholinesterase.	[152]
Papaya	*Carica papaya*	Caricaceae	Fruit	Quercetin, β-sitosterol	It possesses radical scavenging activity.	[153]
Saffron	*Crocus sativus*	Iridaceae	Stigma	Crocin, crocetin, picrocrocin, safranin, and safranal,	It possesses antioxidant and antiamyloid activities.	[154]
Ginger	*Zingiber officinale*	Zingiberaceae	Rhizome	Shagol, gingerol, zingerone	It shows antioxidant properties.	[155]
Sage	*Salvia officinalis*	Lamiaceae	Leaves	Rosmarinic acid, thujone, cineol, and camphor	It shows antioxidant properties. It has cognitive-enhancing properties and helps in preventing age-related problems.	[156]
Camb	*Caryocar brasiliense*	Caryocaracea	Leaf	Gallic acid, quinic acid, quercetin, and quercetin 3-o arabinose	It has neuroprotective effects.	[157]
Coconut	*Cocos nucifera*	Arecaceae	Seed	Caproic acid, Caprylic acid, Capric acid, Lauric acid, and Myristic acid	It helps in preventing Alzheimer’s disease.	[158]
Gouteng	*Uncaria rhynchophylla*	Rubiaceae	Stem	Rhynchophylline, isorhynchophylline, and hirsuteine	It shows free radical scavenging activity and also exhibits protection against kainic acid-induced neuronal damage.	[159]
Aloe vera	*Aloe barbadensis miller*	Aloaceae	Juice	Aloin, β-secretase, aloe-emodin	It improves brain functioning.	[160]
Wuzhuyu	*Tetradium ruticarpum*	Rutaceae	Fruit	Evodiamine, rutaecarpine, evocarpine, and quinoline	It increases the blood flow in the brain and also inhibits the effect of acetylcholinesterase.	[161]
Moringa	*Moring oleifera*	Moringaceae	Leaves	Glycoside niazirin, niaziminim A and B,	It maintains the monoamine level in the brain and helps in treating Alzheimer’s disease.	[162]
Walnut	*Juglans regia*	Juglandaceae	Kernel	α-tocopherol, ellagic acid, and juglone	It reduces the risk of Alzheimer’s disease by reducing oxidative stress and it also shows amyloidogenic activity.	[163,164]
Cinnamon	*Cinnamomum verum*	Lauraceae	Extract of bark	Cinnamaldehyde, eugenol, and trans cinnamaldehyde	It promotes the disassembly of tau filaments and also shows anti-inflammatory activity.	[165]
Tahitian gooseberry	*Phyllanthus acidus*	Phyllanthaceae	Fruit	Terpine	It lowers oxidative stress, decreases lipid peroxidation, and helps in increasing the level of antioxidant enzymes in the brain.	[166]
Fig	*Ficus carica*	Moraceae	Fruit	Quercetin, C-Sitosterol	It has antioxidant activity, exhibits memory-enhancing effects and better learning abilities.	[167]
Pumpkin	*Cucurbita maxima*	Cucurbitaceae	seeds	Ferulic acid, caffeic acid, and coumaric acid	It has antioxidant properties and helps in relieving stress.	[168]
Shankhpushpi	*Convolvulus pluricaulis*	Convolvulaceae	Whole plant	Flavonol glycosides, anthocyanins, and triterpenoids	It is consumed as a tonic for enhancing memory and it calms the nerves.	[169,170]
Strawberry	*Fragaria ananassa*	Rosaceae	Fruit	Pelargonidin	It has antioxidant properties.	[171]
Butterfly pea	*Clitoria ternatea*	Fabaceae	Root and leaf extract	Myricetin, quercetin	It shows antioxidant properties and AChE inhibitor activities.	[172]
Broccoli	*Brassica oleracea* var. *italica*	Brassicaceae	Floret	Kaempferol, sulforaphane	It possesses antioxidant activities and reduces cerebral oedema.	[173]
Spinach	*Spinacia oleracea*	Amaranthaceae	Leaves	Ferulic acid, coumaric acid, quercetin, spinacetin, and myricetin,	It reduces the neuronal death and production of ROS.	[174]
Date palm	*Phoenix dactylifera* L.	Arecaceae	Fruit	Cinnamic acid, caffeic acid, protocatechuic, gallic acid, dactylifiric acid, and epicatechin	It has antioxidant properties and helps in enhancing memory	[175]

## Data Availability

Not applicable.

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
