# Peer review of "Phytochemicals: A Promising Alternative for the Prevention of Alzheimer’s Disease"

_life, 2023, doi:10.3390/life13040999_

Round 1

Reviewer 1 Report

The review article titled “Phytochemicals: a promising alternative for the treatment of Alzheimer's Disease” is well written, and the data taken from the literature were better organized. The novelty of the review article is the collection of factors contributing to Alzheimer's Disease (AD) as well as the factors responsible for the treatment or prevention of AD, however, to date there are several review articles describing the role of phytochemicals or nutraceuticals in the prevention or treatment of AD.  
However, there are some minor suggestions that need to be addressed before the manuscript should be accepted.

1)  In the manuscript the authors mainly focus on the prevention of AD. It would be better if the authors can add the word “prevention” in the title of the article.

2)  Please elaborate more on the clinical trials already done to check the effects of phytochemicals in the treatment or prevention of AD.

3)  Authors have mentioned that they have screened around 360 papers and out of that 245 papers were selected to be included in this review. Please elaborate on what criteria you choose and exclude the articles.

Author Response

1)  In the manuscript the authors mainly focus on the prevention of AD. It would be better if the authors can add the word “prevention” in the title of the article.

 Response: Thank you for your thoughtful suggestion. The change is updated in revised manuscript.

2)  Please elaborate more on the clinical trials already done to check the effects of phytochemicals in the treatment or prevention of AD.

Response: The Information regarding the clinical trials was mention in the previous submitted manuscript. We have incorporated more information in revised manuscript. Thank you.

3)  Authors have mentioned that they have screened around 360 papers and out of that 245 papers were selected to be included in this review. Please elaborate on what criteria you choose and exclude the articles.

Response: Thank you so much for commenting on this. We just need to highlight here for the deep searching in writing this manuscript. Any way we have mention the reason in the abstract that “on the basis of key words and relevant information we selected the papers is mentioned in the revised manuscript.

Reviewer 2 Report

In the present manuscript, the author has presented a detailed review on the use of phytochemicals as a promising alternative for the treatment of Alzheimer’s disease. The author has discussed various factors responsible for the onset of the disease and different treatment options available to either delay the onset or stop further deterioration in the symptoms of patients with AD.  The author has highlighted the merits and demerits of different treatment options and strongly suggests the use of phytochemicals as promising alternative for the treatment of AD. However, I have few questions for the author and want them to address:

1.     There are multiple reviews available on the use of Phytochemicals as a treatment alternatives for the AD, how is the present review different compared to available ones?

https://onlinelibrary.wiley.com/doi/pdf/10.1111/j.1471-4159.2009.06562.x

https://www.hindawi.com/journals/omcl/2012/386527/

https://www.ncbi.nlm.nih.gov/pmc/articles/PMC6274085/.

2.     Most phytochemicals have a pleotropic role in addressing neurological diseases such as Resveratrol, Curcumin etc. I’m curious to know how does the author propose to use these phytochemicals as treatment alternative?

Author Response

Reviewer #2

In the present manuscript, the author has presented a detailed review on the use of phytochemicals as a promising alternative for the treatment of Alzheimer’s disease. The author has discussed various factors responsible for the onset of the disease and different treatment options available to either delay the onset or stop further deterioration in the symptoms of patients with AD.  The author has highlighted the merits and demerits of different treatment options and strongly suggests the use of phytochemicals as promising alternative for the treatment of AD.

Response: Thank you so much for summarizing our work and taking time to review our manuscript.

However, I have few questions for the author and want them to address:

  • There are multiple reviews available on the use of Phytochemicals as a treatment alternative for the AD, how is the present review different compared to available ones?

https://onlinelibrary.wiley.com/doi/pdf/10.1111/j.1471-4159.2009.06562.x

https://www.hindawi.com/journals/omcl/2012/386527/

https://www.ncbi.nlm.nih.gov/pmc/articles/PMC6274085/.

Response: Thanks for your nice comments. I agree with the reviewer that there were many papers on phytochemicals but the content we presented here in the manuscript were different. I hope the reviewers could agree on this fact and we have mentioned the recent reviews in our manuscript so that the common reader of our manuscript could be benefitted.  The other reason would be that the above-mentioned papers did not explain the factors contributing to the onset of Alzheimer’s disease and also chemical based treatment was also not explained. In our paper we have tried to compare plant-based treatment with the available chemical-based treatment.

2)  Most phytochemicals have a pleotropic role in addressing neurological diseases such as Resveratrol, Curcumin etc. I’m curious to know how does the author propose to use these phytochemicals as treatment alternative?

  Response:  Thank you so much for your thoughtful comments. We agree with the reviewers comments that these phytochemicals that were mentioned in the manuscript showed pleotropic effect. Not only they used to treat neurodegenerative diseases, they are used in treatment of several metabolic disorders such as diabetes, obesity and CVD. We think that these polyphenols have rich sources of antioxidant anti-inflammatory properties and that’s why drug derived from these phytochemical used for various purpose. Concerning our cited paper used we pointed out “It has been reported that resveratrol when taken in combination with the available drug for AD i.e., donepezil works better in reducing AD symptoms. Curcumin with pleotropic activity target many pathological mechanisms of AD (Aβ plaque formation, tau hyperphosphorylation etc.) as it possesses strong anti-Aβ, anti-inflammatory and antioxidant properties”.

   https://doi.org/10.1007/s13205-021-02879-5

   https://doi.org/10.1155/2022/9148650   

   https://doi.org/10.2174/1570159X19666210823103020

Reviewer 3 Report

This manuscript provides an overview on medicinal plants and phytochemicals for potential treatment of Alzheimer's disease (AD), a neurological condition that worsens with ageing and affects memory and cognitive function. This review complies the references related to AD treatment, both plants and the active compounds.  

Overall, the manuscript is well written, and provides useful information for readers. In order to improve this manuscript, please consider the comments and suggestions, which are listed below.

1.     Keywords should include the word “Medicinal plants”

2.     Other recent reviews on the same or similar topics should be mentioned in the introduction and compared with this review, and explain why we need this manuscript as a review article.

3.     “There are several environmental factors linked to AD, although the majority of research focuses on three of them: exposure to aluminium (Al), the impact of head injury, and the influence of food and malnutrition.”; please add references to this fact.

4.     “2.8. Infections”; recently a few works gave hypothesis that COVID-19 may cause AD. Please try to find these works and put in the content. This would keep this review up-to-date information. For example, Association of COVID-19 with New-Onset Alzheimer's Disease, J Alzheimers Dis. 2022;89(2):411-414. doi: 10.3233/JAD-220717. There are more papers on this aspect, please search for the information. It would also be interesting to mention this new discovery in the abstract, which is attractive to readers.

5.     “There are already 24 million cases of AD documented globally, and by 2050, the overall number of AD patients is expected to become more than triple.”; please provide references.

6.     Please revise “3.1.2. N-methyl D-aspartate receptor antagonists” to “3.1.2. N-methyl D-aspartate receptor (NMDAR) antagonists”.

7.     7. Future prospective”; any suggestion or recommendation for COVID-19 that may cause AD? Any roles of medicinal plants?

Author Response

Comments and Suggestions for Authors

This manuscript provides an overview on medicinal plants and phytochemicals for potential treatment of Alzheimer's disease (AD), a neurological condition that worsens with ageing and affects memory and cognitive function. This review complies the references related to AD treatment, both plants and the active compounds.  

Overall, the manuscript is well written, and provides useful information for readers. In order to improve this manuscript, please consider the comments and suggestions, which are listed below.

1)     Keywords should include the word “Medicinal plants”

Response: The suggestion is incorporated in the revised manuscript.

2)   Other recent reviews on the same or similar topics should be mentioned in the introduction and compared with this review, and explain why we need this manuscript as a review article.

   Response: We agree with the reviewers’ comments. However, we have updated several information which were highlighted in the manuscript and cited new references for the common reader of this manuscript. Moreover, we have added important points on Phyto chemicals that are not completely present in other reviews. Other specific reason are highlighted in the last paragraph of the revised manuscript.

3) “There are several environmental factors linked to AD, although the majority of research focuses on three of them: exposure to aluminium (Al), the impact of head injury, and the influence of food and malnutrition.”; please add references to this fact.

Response: Reference is added in the revised manuscript.

4) “2.8. Infections”; recently a few works gave hypothesis that COVID-19 may cause AD. Please try to find these works and put in the content. This would keep this review up-to-date information. For example, Association of COVID-19 with New-Onset Alzheimer's Disease, J Alzheimers Dis. 2022;89(2):411-414. doi: 10.3233/JAD-220717. There are more papers on this aspect, please search for the information. It would also be interesting to mention this new discovery in the abstract, which is attractive to readers.

Response: The revised manuscript is updated with the above-mentioned cause of Alzheimer’s disease.

5) “There are already 24 million cases of AD documented globally, and by 2050, the overall number of AD patients is expected to become more than triple.”; please provide references.

Response: Reference is added in the revised manuscript. Thank you so much

6)   Please revise “3.1.2. N-methyl D-aspartate receptor antagonists” to “3.1.2. N-methyl D-aspartate receptor (NMDAR) antagonists”.

Response: The change is updated in the revised manuscript.

7) “7. Future prospective”; any suggestion or recommendation for COVID-19 that may cause AD? Any roles of medicinal plants?

Response: The suggestion is incorporated in the revised manuscript. Thank you so much for taking your time in providing the thoughtful comments. Please let us know if any further suggestion required. Thank you.

Reviewer 4 Report

In the present manuscript the authors review  the phytochemical constituents of different plants that are used for the treatment of AD. The manuscript is well written and organized. The figures’ quality is of a high standard while the tables are clear and informative. The review is well-structured starting with a solid introduction, analyzing the AD factors and treatments. The plant description is developed in a  smart and efficient style. Each plan is described by a few sentences where all the most important scientific features are listed and explained. The number and quality of refences is appropriate for a comprehensive review. I have enjoyed reading the review and I am glad to recommend publication in the present form.

Author Response

In the present manuscript the authors review the phytochemical constituents of different plants that are used for the treatment of AD. The manuscript is well written and organized. The figures’ quality is of a high standard while the tables are clear and informative. The review is well-structured starting with a solid introduction, analyzing the AD factors and treatments. The plant description is developed in a smart and efficient style. Each plan is described by a few sentences where all the most important scientific features are listed and explained. The number and quality of refences is appropriate for a comprehensive review. I have enjoyed reading the review and I am glad to recommend publication in the present form.

Response: Thank you for your positive comments.